# Longitudinal Changes in Sitting Patterns, Physical Activity, and Health Outcomes in Adolescents

**DOI:** 10.3390/children6010002

**Published:** 2018-12-23

**Authors:** Ana María Contardo Ayala, Jo Salmon, David W. Dunstan, Lauren Arundell, Kate Parker, Anna Timperio

**Affiliations:** 1Institute for Physical Activity and Nutrition (IPAN), School of Exercise and Nutrition Sciences, Deakin University, Geelong 3220, Australia, jo.salmon@deakin.edu.au (J.S.); David.Dunstan@bakeridi.edu.au (D.W.D.); lauren.arundell@deakin.edu.au (L.A.); k.parker@deakin.edu.au (K.P.); anna.timperio@deakin.edu.au (A.T.); 2Physical Activity Laboratory, Baker Heart and Diabetes Institute, Melbourne 3004, Australia; 3Mary MacKillop Institute for Health Research, Australian Catholic University, Melbourne 3000, Australia

**Keywords:** sitting time, body mass index, waist circumference, activPAL, ActiGraph, adolescents

## Abstract

This study examined two-year changes in patterns of activity and associations with body mass index (BMI) and waist circumference (WC) among adolescents. Inclinometers (activPAL) assessed sitting, sitting bouts, standing, stepping, and breaks from sitting. ActiGraph-accelerometers assessed sedentary time (SED), light-intensity physical activity (LIPA, stratified as low- and high-LIPA), and moderate-to-vigorous physical activity (MVPA). Anthropometric measures were objectively assessed at baseline and self-reported at follow-up. Data from 324 and 67 participants were obtained at baseline and follow-up, respectively. Multilevel mixed-effects linear regression models examined changes over time, and associations between baseline values and BMI and WC at follow-up. There were significant increases in BMI (0.6 kg/m^2^) and durations of prolonged sitting (26.4 min/day) and SED (52 min/day), and significant decreases in stepping (−19 min/day), LIPA (−33 min/day), low-LIPA (−26 min/day), high-LIPA (−6.3 min/day), MVPA (−19 min/day), and the number of breaks/day (−8). High baseline sitting time was associated (*p* = 0.086) with higher BMI at follow-up. There were no significant associations between baseline sitting, prolonged sitting, LIPA, or MVPA with WC. Although changes in daily activity patterns were not in a favourable direction, there were no clear associations with BMI or WC. Research with larger sample sizes and more time points is needed.

## 1. Introduction

Adolescents spend most of their waking hours in sedentary pursuits (i.e., behaviours characterised by low energy expenditure, <1.5 metabolic equivalent units [METS], while in a sitting, reclining or lying posture during waking hours [1]). In addition, in Australia, nine out of ten adolescents do not meet the minimum recommendation of 60 min of moderate-to-vigorous physical activity (MVPA) every day [2]. Other developed countries have also reported low levels of physical activity and extensive time spent in sedentary behaviours [3]. Adolescents may be particularly susceptible to the negative health impacts associated with these behaviours [4].

Longitudinal evidence indicates that accelerometer-measured sedentary time (SED) increases by approximately 23 to 30 min/year [5,6,7,8,9,10,11] and sedentary bouts become longer and less interrupted [10,12,13] between childhood and adolescence. There is also evidence suggesting that sedentary behaviour tracks moderately from childhood to young adulthood [14]. While both MVPA and light-intensity physical activity (LIPA, e.g., standing, slow walk) decrease with age [12,15], the absolute decrease in daily MVPA is less than that in LIPA, most likely because adolescents spend little time in MVPA (5–7% of daily waking hours) [16,17]. Therefore, decreases in adolescents’ LIPA may reflect displacement by time spent sedentary [18]. However, few studies have reported longitudinal changes in LIPA and those that have only considered the overall range of LIPA (i.e., standing to slow walking), despite recent evidence suggesting different health benefits at the low (low-LIPA) and high (high-LIPA) spectrum of LIPA [19,20]. In addition, there is limited evidence reporting longitudinal changes in objectively-measured sitting time using inclinometers (which provide postural information). Most inclinometer studies have examined changes in total volume of activity at each intensity without considering changes in patterns, such as breaks from sitting or time spent in continuous sitting. It is important to identify how such patterns change over time as evidence suggests [21,22] that the manner in which sitting time is accumulated can impact adult health. However, longitudinal associations between sitting patterns and health remain largely unexplored among young people.

Evidence indicates that excessive SED time is associated with adverse cardio-metabolic outcomes among adults [23]; however, this relationship is inconsistent among adolescents [24]. Longitudinal research has shown that increased SED is positively associated with body fatness, but not with body mass index (BMI), over an 8-year period (age 7–15 years) [25], and associated with greater increases in BMI over a 6-year period (age 9–15 years) [26]. However, SED was not associated with changes in adiposity markers in other studies [27,28]. Shorter sedentary bouts have been associated with decreases in adiposity (i.e., fat mass index), between nine and 12 years of age [25]. All of these studies have used accelerometers to assess SED time, therefore, the longitudinal relationship between objectively-assessed sitting, sitting patterns and adolescents’ health is unknown.

Conversely, there is consistent longitudinal evidence of inverse relationships between accelerometer-assessed MVPA and health outcomes in school-aged children and adolescents [29]. Among children, MVPA has been prospectively inversely associated with adiposity (i.e., trunk fat mass, body fat mass and fat mass index [30], and cardio-metabolic risk factors (i.e., blood pressure, triglycerides, homeostatic model assessment for insulin resistance (HOMA)) [12], suggesting that active children are less likely to be obese in early adolescence [31]. In adolescents, the longitudinal health effects of LIPA remain unclear; in children, LIPA is negatively associated with fat mass [32], while among adolescents it is positively associated with adiposity markers [33]. It is important to consider that these studies measured LIPA with accelerometers which are unable to detect differences between sitting and standing, and potentially misclassify standing still (i.e., low-LIPA) as sedentary [34].

To our knowledge, few studies have examined longitudinal changes in sitting and SED time, LIPA, MVPA, and activity patterns over time among adolescents. Furthermore, associations between objectively-assessed LIPA and adiposity markers among adolescents have been infrequently examined, and no studies have examined associations between objectively-measured sitting and sitting patterns and adolescents’ adiposity. The aim of the present study was to examine changes in inclinometer—and accelerometer-measured sitting, sitting patterns (i.e., continuous sitting bouts and breaks from sitting), SED time, LIPA, MVPA and adiposity markers (i.e., BMI and waist circumference, WC) over time; and to determine the associations between sitting and activity at baseline with adiposity markers over two years among a sample of Australian adolescents.

## 2. Materials and Methods

For this study, baseline data were collected as part of The Neighbourhood Activity in Youth Project: The NEArbY Study, and a subset of participants was followed up after two years. The Deakin University Human Ethics Advisory Group (HEAG-H 152_2013) and pertinent education authorities approved the study.

### 2.1. Recruitment

#### 2.1.1. Baseline

Recruitment of a diverse sample of adolescents from high/low walkable and income areas has been described previously [35]. In brief, secondary schools in Melbourne were approached and invited to participate. Schools could choose which year levels (grades) to involve in the study. Once school consent was received (*n* = 18), research staff delivered a short 15-min presentation to eligible students (i.e., explaining what was involved for participants and parents) and distributed packs containing information about the study for the adolescents and their parent, a consent form for parents to sign on behalf of their child, and a parent survey (with its own consent form). Overall, 1454 recruitment packs were delivered within the 18 schools. In total, 528 consent forms were returned. The parent survey included a question where parents could opt out of future contact from the research team.

#### 2.1.2. Follow-Up

Recruitment occurred two years after baseline involvement. At baseline, there were 376 participants whose parents did not opt-out of future contact (52 of whom were uncontactable). A total of 324 adolescents who were contactable formed the sampling frame for the follow-up study. Participants whose parents declined future follow-up were significantly older at baseline compared with the follow-up sampling frame, but no differences were found for the anthropometric or sedentary and activity variables between these groups. A newsletter with brief findings from the baseline NEArbY study (e.g., percentage of participants that owned a dog, lived near a park and watched TV in their room) and a notification about the follow-up study were sent to participants. Two weeks later, an invitation letter, a new plain language statement and a consent form were mailed to participants. Adolescents under 18 years of age required consent from a parent/carer/guardian to participate. Compensation in the form of a gift voucher was offered to all participants in appreciation of their time. Two reminders to complete and return the consent form were sent to non-responders one week apart via short message service (SMS; if supplied at baseline), telephone or mail. In total, 67 returned consent forms (21% response rate). Participants that did not respond at follow-up, were significantly older at baseline compared with participants that consented for follow-up assessment, but no differences were found for anthropometric, sedentary or activity variables.

### 2.2. Data Collection

#### 2.2.1. Baseline

Adolescents completed an online survey at school and wore an ActiGraph accelerometer for one week. Parents indicated on the consent form if they also gave permission for their child to wear an activPAL inclinometer for one week and have their height, body mass and waist circumference measured. Data collection occurred between August 2014 and December 2015.

#### 2.2.2. Follow-Up

Adolescents completed an online survey in their own time and wore an accelerometer for one week. They could also consent to wear an activPAL inclinometer. After the participant consent form was received, participants were sent their activity monitors and a measuring tape for self-assessment of waist circumference, and an electronic link to the online survey was emailed. Monitors were collected from the participants’ home seven days later. Data collection occurred between August 2016 and December 2017.

### 2.3. Measures

Demographics: Adolescents self-reported their age and sex.

#### 2.3.1. Activity Monitors

At baseline and follow-up, participants wore an activPAL3C inclinometer (PAL Technologies Ltd., Glasgow, UK) at the mid-point of their right thigh (on the front), attached via an elastic garter band. This monitor identifies limb position and is a valid and reliable device for measuring sitting, standing and stepping time in youth [36,37]. Participants also wore an ActiGraph GT3X accelerometer (Actigraph LLC, Pensacola, FL, USA) on a belt on their right hip and were instructed to wear both monitors during all waking hours except during water-based activities. All activPAL and ActiGraph data were downloaded using manufacturer registered software (ActivPAL Professional v7.2.29, PAL Technologies Ltd., and ActiLife v6.11.8, Actigraph LLC respectively), in 15-s epochs and processed using a customised Microsoft excel macro. Non-wear time was defined as 60 min of consecutives zero counts for both devices [38]. A valid day was defined as ≥8 h/day of wear time for week days and ≥7 h/day for weekend days. Adolescents were included in the analysis if they had worn the monitor for at least four valid days (week and/or weekend days).

Average minutes per day spent sitting, in continuous sitting bouts (>5, >10, >20 min duration, without interruptions allowed), standing, stepping and the number of transitions from sitting to standing (i.e., breaks from sitting) were computed from the activPAL data. Freedson youth age-adjusted cut-points [39] were applied to accelerometer data to determine average minutes/day of SED (<100 counts/minute), LIPA (101–3.99 METS), low-LIPA (101–799 counts/min), high-LIPA (800 counts/min–3.99 METS) and MVPA (≥4 METS, i.e., for a 15 year old it is 2781 counts/min). The low-LIPA cut-point of up to 799 counts/min was based on previous research suggesting this threshold typically captures static LIPA such as standing [20]. For 17 participants aged ≥18 years at follow-up, Freedson adult cut-points were used [40].

#### 2.3.2. Baseline Anthropometrics Measures

Adolescents’ anthropometric assessments were taken by trained project staff. Height was measured to the nearest 0.1 cm using a portable stadiometer (Seca217, Seca, Hamburg, Germany). *Body mass* was measured to the nearest 0.1 kg using a portable electronic scale (Tanita BC-351, Tanita, Tokyo, Japan). Duplicate measurements were taken and when a discrepancy of over 0.5 cm or 0.5 kg was noted, a third measurement was taken and an average calculated. All measurements were taken in a private area in school uniform and without shoes. Waist circumference was measured using a flexible measuring tape. Adolescents were asked to remove any bulky clothing (e.g., jackets and jumpers) and pull their top/dress tight around their waist. This allowed research staff to identify the umbilicus point, in the midaxillary plane. The average of two waist circumference measures was used and where there was a discrepancy greater than 1 cm, a third measurement was taken and an average calculated.

#### 2.3.3. Follow-up Anthropometrics Measures

Adolescents’ height and weight were self-reported via the online survey. To aid the self-reported waist circumference, participants were asked to use the tape measure provided and to follow written instructions. Studies show small discrepancies between objective and self-measured height, weight and waist circumference in adults and young people (e.g., underestimation of weight) [41,42]; self-measured adiposity has been used in epidemiological studies when objective measured are not possible. Body mass index (BMI = kg/m^2^) was calculated and categorised according to the International Obesity Task Force definitions of healthy weight or overweight/obese (i.e., based on age and sex), at baseline and follow-up [43].

### 2.4. Statistical Analysis

All statistical analyses were conducted using Stata 15.0 (StataCorp LP., College Station. TX, USA). Statistical significance was set at *p* <0.05. Before the analysis all activPAL and ActiGraph-derived outcomes variables were standardised according to total wear time as follows: ((duration of X within waking hours/wear time within waking hours) multiplied by 960 min), where X is the activity (e.g., sitting, LIPA and MVPA) and waking hours are equivalent to 960 min (16 h).

Descriptive statistics (independent *t*-test, Pearson’s chi-squared) were used to compare data from the two time points. The daily percentage of time spent in sitting, LIPA (total, low, high) and MVPA was calculated considering adolescents waking hours (16 h/day, from 06:00 to 22:00). Multilevel mixed-effects linear regression models were used (where analysis unit, individuals, were nested within baseline schools) to examine two-year changes in the time spent in sitting, sitting bouts (>5, >10, >20 min duration), standing, stepping, breaks from sitting (number), SED, LIPA (total, low, high), MVPA, and in BMI and WC, and effect sizes were calculated (Cohen’s *d*). These models maximise data by conservatively allowing missing follow-up data. Additional multilevel regression models were used to determine the association between changes in sedentary/sitting and activity measures over time and changes on BMI. To determine prospective associations between sitting, sitting bouts, standing, stepping, LIPA, low-LIPA, high-LIPA and MVPA at baseline and two-year changes in BMI and waist circumference, interactions with time were examined in separate multilevel mixed-effects linear regression models. Significant interactions (*p* = 0.05) were further explored (graphically) by describing the association at different baseline levels: at the mean, at one standard deviation below and one standard above the mean (e.g., sitting, sitting bouts, standing, stepping, LIPA, low-LIPA, high-LIPA and MVPA).

## 3. Results

Of the 376 adolescents in the sampling frame, 366 wore an ActiGraph (valid data = 280) at baseline and 67 (valid data = 51) at follow-up. For the activPAL, 336 participants provided some data (valid data = 234) at baseline and 62 (valid data = 51) at follow-up. Anthropometric measures were obtained from 254 participants at baseline and 60 at follow-up.

Participants (59% girls) were aged 14.9 ± 1.6 years at baseline and 16.4 ± 1.4 at follow-up. The proportion of participants categorised as healthy weight was 72% at baseline and 64% at follow-up. On average, during waking hours, participants spent 68% and 69% of their non-sleep time sitting (of which 55% and 57% was spent on prolonged sitting >20 min), 27% and 24% on LIPA and 5% and 4% on MVPA, at baseline and follow-up, respectively.

Table 1 shows that after two years, participants’ BMI were significantly higher and there was a significant increase in time spent in prolonged (>20 min) bouts of sitting and in SED. Adolescents spent significantly less time stepping, had fewer breaks from sitting, and spent less time in LIPA (low-LIPA and high-LIPA) and MVPA.

Further analysis showed that the reduction in the number of breaks from sitting over time was associated with an increase in BMI (b = −0.03, 95% CI: −0.05, −0.00, p = 0.011). There were no significant interactions between adolescents’ baseline sedentary or physical activity variables and patterns and time in relation to their waist circumference or BMI, with the exception of a close to significant interaction (p = 0.086) between baseline duration of sitting and time for BMI. Further exploration (Figure 1) showed that highest sitting levels at baseline (1 standard deviation above the mean) had a steeper increase in BMI over time.

## 4. Discussion

This study examined changes in sitting, sitting patterns, physical activity, and adiposity markers over time and associations between sitting and activity at baseline and adiposity markers over two years amongst Australian adolescents. Overall, participants’ BMI, sedentary time and prolonged sitting bouts increased, while total physical activity and breaks from sitting decreased over the two-year period. No significant interactions were found between baseline SED, sitting patterns and physical activity variables and time for WC or BMI, except for adolescents’ baseline sitting and BMI (approached significance). Further probing indicated that those who spent the most time sitting at baseline had higher BMI after two years.

To our knowledge, this is the first study to describe longitudinal changes in objectively measured sitting time and patterns among adolescents. The non-significant increase in total average daily sitting time could be attributed to the very high levels of sitting at baseline (68% of total daily waking hours), which is consistent with previous research [44,45]. The way sitting time was accumulated changed over time, with an increase in time spent in prolonged sitting bouts of >20 min and a decline in the number of breaks from sitting. In addition, fewer breaks from sitting were negatively associated with BMI over time. In adults, prolonged and uninterrupted sitting have been associated with adverse health outcomes [46], on the other hand, breaks from sitting have been linked to health benefits [47,48]. Based on the current results, adolescents may be at an elevated risk due to their high total sitting time, and their sitting patterns, especially since those who spent the most time sitting at baseline had the sharpest increase in BMI over the two years. Future research should not only examine the impact of high total sitting time on health, but also the impact of longer sitting bouts and fewer breaks on health risk markers. The significant changes in total SED time from the ActiGraph over time, but not total sitting, could be attributed to the fact that the ActiGraph accelerometer does not assess posture directly and may include LIPA (e.g., standing not stepping) as sedentary time [37], overestimating this increase.

Time spent in LIPA decreased, which is consistent with the similar changes in stepping time from the activPAL and low-LIPA measured by accelerometry. The large reduction in low-LIPA (e.g., standing still, very light movement) accounts for the greater decrease in LIPA. To our knowledge, no studies have examined longitudinal changes in the sublevels of high- and low-LIPA among adolescents. This is an important consideration due the high proportion of waking hours spent in this intensity level. Similar to previous studies [12], MVPA also decreased and reached only 35 min/day at follow-up, indicating that a large proportion of adolescents in this sample did not meet physical activity recommendations of 60 min of MVPA every day, and therefore could be predisposed to an unhealthy activity profile.

There were no significant prospective associations between baseline time spent in each of the sitting, SED or physical activity variables and patterns in relation to WC. This is in contrary to previous literature that showed prospective inverse relationships between MVPA and WC [12]; however, typically these studies have focused only on children. In this study the relationship between high baseline sitting and higher BMI after two years approached significance, suggesting that excessive sitting time may have negative impact on health in the long term. To our knowledge, this is the only study that investigated the prospective associations of sitting and sitting patterns as well as associations between LIPA (low- and high-LIPA) and adiposity markers. Further longitudinal studies, including those with a longer duration, larger sample sizes and analyses that consider the complex relationships between health and SED and physical activity (e.g., compositional analysis [49]), are necessary to elucidate the impact of these behaviours on health.

The main strengths of this study include the diverse sample recruited from across Melbourne and the use of objective measures of sitting time (i.e., inclinometers) and physical activity (i.e., accelerometers), which allowed the analysis of sitting and activity patterns. This study also had some limitations, including the small follow-up sample (due to the low response rate) and the lack of objective measures of adiposity markers at follow-up, suggesting that these results should be interpreted with caution. Future research should consider the limitations of this study to further elucidate the long-term health impact of sitting and activity behaviours.

## 5. Conclusions

Changes in average daily sitting and activity intensities and patterns among adolescents followed the direction considered less desirable for health outcomes. However, there were no clear prospective associations with WC, but there was a trend with baseline sitting duration and BMI two years later. In concordance with previous evidence, the findings from this study suggest that adolescents may be at risk of negative health impacts associated with decreases in physical activity and increases in prolonged sitting over time. Longitudinal research with larger sample sizes and additional assessments points is needed.

## Figures and Tables

**Figure 1 children-06-00002-f001:**
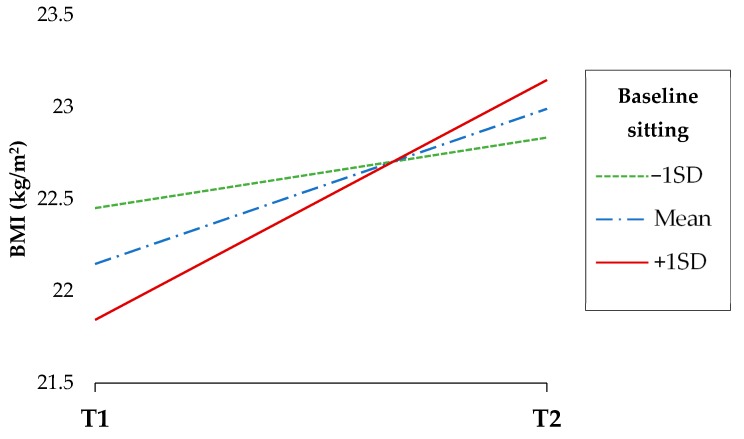
Prospective association of baseline sitting time (at the mean, and at one standard deviation below and above mean) on BMI. T1: Baseline; T2; follow-up; SD: standard deviation; BMI = body mass index.

**Table 1 children-06-00002-t001:** Participants’ characteristics and 2-year changes in sedentary/sitting and activity measures and adiposity markers and effect sizes.

	Baseline	2 Year Follow-Up	Change from Baseline		
	Mean (SD)	Mean (SD)	*b* (95% CI)	*p*	Cohen’s *d*
**WC** (cm)	77.1 ± 9.98	79.9 ± 16.9	1.9 (−1.0, 4.8)	0.200	0.277
**BMI** (kg/m^2^)	22.2 ± 3.7	23.2 ± 3.8	0.6 (0.1, 01.1)	**0.030**	**0.542**
**Activity Measures**					
**activPAL**	**min/day (SD)**	**min/day (SD)**			
Sitting	649.1 ± 91.9	663.7 ± 116.8	18.0 (−9.9, 46)	0.200	0.226
SIT bouts >5 min	500.4 ± 93.3	506.4 ± 100.7	5.6 (−21.62, 32.8)	0.678	0.087
SIT bouts >10 min	448 ± 91.8	461.7 ± 103.3	14.6 (−12.7, 41.9)	0.287	0.211
SIT bouts >20 min	354.8 ± 84.2	378.35 ± 106.1	26.4 (1.0, 51.8)	**0.042**	0.425
Breaks from sitting (n/day)	53.4 ± 16.8	47.3 ± 15.5	−7.8 (−12.5, −3.1)	**0.002**	**0.888**
Standing	194.9 ± 71.2	196.5± 77.3	1.8 (−17.9, 21.6)	0.851	0.459
Stepping	116.2 ± 41.9	100.0 ± 45.6	−19.1 (−31.7, −6.5)	**0.004**	0.466
**Actigraph**					
SED	648.2 ± 65.1	696.8 ± 86.1	51.8 (37.1, 66.5)	**0.000**	**0.948**
LIPA	260.2 ± 54.5	228.1 ± 73.7	−32.5 (−45.2, −19.7)	**0.000**	**0.701**
Low-LIPA	166.3 ± 35.5	141 ± 40.4	−26.2 (−34.1, −18.2)	**0.000**	**0.845**
High-LIPA	93.9 ± 25.2	87.1 ± 38.4	−6.3 (−12.6, −0.04)	**0.048**	0.317
MVPA	51.6 ± 26.7	35.1 ± 28.5	−19.4 (−25.5, −13.4)	**0.000**	**0.715**

Abbreviations: CI = confidence interval; SD = standard deviation; WC = waist circumference; BMI = body mass index; SIT = sitting bouts; n = number; SED = Sedentary time; LIPA = light intensity physical activity; low-LIPA = low light intensity physical activity; high-LIPA = high light intensity physical activity; MVPA = moderate-to-vigorous physical activity. Significant differences (*p* <0.05) are highlighted in **bold**.

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
