# Peer review of "Longitudinal Changes in Sitting Patterns, Physical Activity, and Health Outcomes in Adolescents"

_children, 2018, doi:10.3390/children6010002_

Round 1
Reviewer 1 Report
General comments:
The purpose of this study was to examine associations between objectively measured sitting and physical activity measured at baseline with change in adiposity markers at a two-year follow-up in adolescents. The study is timely, well written and easy to follow. However, there are some methodological limitations, including low response rate at follow-up, and not reporting the association between changesin sitting and physical activity variables with changesin adiposity markers. Detailed comments are outlined below.
Introduction:
Lines 36-37: Please provide a reference to support this statement. Why adolescents would be particularly susceptible, as compared to other age groups?
Methods:
Line 91: Please elaborate what kind of information was given about the study. Was the information about the present study or the previous study (ref. 35). Moreover, please give more details on what findings were presented at follow-up recruitment (line 102). This may affect possible selection bias.
Line 94: Please report how many consent forms were delivered.
Line 107. Please give more details how the response rate was calculated (67 / 376 = 18%, not 21%).
Line 107. The response rate is low. Please report possible differences at baseline between responders and non-responders at follow-up.
Line 142. Were the same cut-points used at baseline and follow-up for all participants?
Line 186. Please clarify what an “acceptable weight” means?
Line 204. Please modify: “baseline duration of sitting and time on BMI.”
Line 207. It would be interesting to see associations between changes in sitting and physical activity variables with changes in adiposity markers. Why were these left out? Would it be possible to include these in the paper?
Lines 225-227. Could the low response rate and selection bias at follow up affect this (i.e. the most sedentary ones at baseline dropped out before follow-up)?
Line 246. Please refer to the specific physical activity recommendations.
Line 252. Please correct: “To our knowledge, this the only study”.
Line 269. Please be specific that the sitting time was measured at baseline, and the association does not consider changes in sitting time.
Author Response
Dear Reviewer 1:
We greatly appreciate the opportunity to revise and resubmit our manuscript for consideration for publication by Children. We found the feedback thoughtful and constructive, and believe that we have been able to significantly strengthen the manuscript.
Please see the attached the response to the reviewers.
Regards,
Ana María Contardo Ayala, PT MSc.
PhD Candidate
Institute for Physical Activity and Nutrition (IPAN)
School of Exercise and Nutrition Sciences
Deakin University
The purpose of this study was to examine associations between objectively measured sitting and physical activity measured at baseline with change in adiposity markers at a two-year follow-up in adolescents. The study is timely, well written and easy to follow. However, there are some methodological limitations, including low response rate at follow-up, and not reporting the association between changes in sitting and physical activity variables with changes in adiposity markers. Detailed comments are outlined below.
Response: Thank you for your comments on our manuscript.
Introduction:
1. Lines 36-37: Please provide a reference to support this statement. Why adolescents would be particularly susceptible, as compared to other age groups?
Response: A reference has been added as suggested to the text.
Ref 4: Tarp, J.; Brond, J.C.; Andersen, L.B.; Moller, N.C.; Froberg, K.; Grontved, A. Physical activity, sedentary behavior, and long-term cardiovascular risk in young people: A review and discussion of methodology in prospective studies. J Sport Health Sci 2016, 5, 145-150, doi:10.1016/j.jshs.2016.03.004
Methods:
2. Line 91: Please elaborate what kind of information was given about the study. Was the information about the present study or the previous study (ref. 35). Moreover, please give more details on what findings were presented at follow-up recruitment (line 102). This may affect possible selection bias
Response: Thank you to the reviewer for this suggestion. We have added the brackets to clarify the information was given about the study:
Line 94-95: “…(i.e. explaining what was involved for participants and parents)”.
To clarify, the findings presented from the NEArbY baseline study in the follow-up recruitment presented statistics unrelated to the current study and are highly unlikely to result in selection bias. For example, findings included the percentage of participants that have a dog and walked it, lived near a park, had access to a mobile phone, consumed fruit and vegetables, watched TV on their room and had access to a laptop. To clarify this point we have added the following between brackets:
Lines 102-103: “…(e.g. percentage of participants that owned a dog, lived near a park and watched TV in their room)”
3. Line 94: Please report how many consent forms were delivered
Response: Overall, 1454 recruitment packs were delivered within the 18 schools, however repeat visits were made to some schools and the number of recruitment packs delivered during one visit to one of the schools was not recorded. Therefore we are unable to report the final number of consent forms delivered. The average number of recruitment packs (n=1454) has been added in the line 97
Line 97: Overall, 1454 recruitment packs were delivered within the 18 schools.
4. Line 107. Please give more details how the response rate was calculated (67 / 376 = 18%, not 21%)
Response: The response rate was calculated with the 324 participants whose parents did not opt-out and were contactable (376-52), as described in the lines 99-101. This has now been clarified.
Lines 100-101: “At baseline, there were 376 participants whose parents did not opt-out of future contact (52 of whom were uncontactable). A total of 324 parents who were contactable…”
5. Line 107. The response rate is low. Please report possible differences at baseline between responders and non-responders at follow-up
Response: We have now reported on differences in participants who did and did not respond at follow-up. The following sentence has been added to the text:
Lines 113-116: “Participants that did not respond at follow-up, were significantly older at baseline compared with participants that consented for follow-up assessment, but no differences were found for anthropometric, sedentary or activity variables.
6. Line 142. Were the same cut-points used at baseline and follow-up for all participants?
Response: Yes, the same protocols, cut-points (i.e.age-adjusted) and validity criteria were used at baseline and follow-up. To clarify this, the following was added:
Line 132: "At baseline and follow-up, participants…”
7. Line 186. Please clarify what an “acceptable weight” means?
Response: The definition of acceptable weight or healthy weight is based on the International Obesity Task Force definitions (based on sex and age). To be consistent through the text, “acceptable weight” has been replaced with “healthy weight” (line 205). In addition, the following was added in the line 174: (i.e. based on age and sex).
8. Line 204. Please modify: “baseline duration of sitting and time on BMI.
Response: Thank you to the reviewer for this suggestion. The sentence is now written as follows:
“…baseline duration of sitting and time for BMI”.
9. Line 207. It would be interesting to see associations between changes in sitting and physical activity variables with changes in adiposity markers. Why were these left out? Would it be possible to include these in the paper?
Response: We originally did not include change scores in the activity variables as the coefficient can be challenging to interpret. However, we have now included associations between changes in sitting/sedentary and activity variables and changes on BMI over time as requested. Waist circumference did not significantly change over time, therefore, we excluded that outcome variable from the additional analyses.
To address this suggestion, part on the manuscript has been modified and new text has been added as follow:
Lines 189-191: “Additional multilevel regression models were used to determine the association between changes in sedentary/sitting and activity measures over time and changes on BMI.
Lines 221-222: “Further analysis showed that the reduction in the number of break from sitting over time was associated with an increase in BMI (b= -0.03, 95% CI: -0.05, -0.00, p=0.011)
Lines 250-251:“In addition, fewer in the number of breaks from sitting were negatively associated with BMI over time”.
10. Lines 225-227. Could the low response rate and selection bias at follow up affect this (i.e. the most sedentary ones at baseline dropped out before follow-up)?
Response: Thank you for your comment. As noted in point number 5 above, there were no significant differences in anthropometrics, sedentary or physical activity variables between respondents and non-respondents at follow-up (which has now been added to the text lines 112-115). However, the small number of participants at follow-up most likely affected the power to detect relationships between the sitting and physical activity variables and health outcomes.
11. Line 246. Please refer to the specific physical activity recommendations
Response: The following text has now been added to line 268:
“…of 60 minutes of MVPA every day”
12. Line 252. Please correct: “To our knowledge, this the only study”
Response: Thank you for this correction. The phrase now reads as follow: “To our knowledge, this is the only study” (line 276)
n
13. Line 269. Please be specific that the sitting time was measured at baseline, and the association does not consider changes in sitting time.
Response: Thank you for this suggestion. The word baseline was added to the sentence as follow:
“…but there was a trend with baseline sitting duration and BMI over two years.”
Reviewer 2 Report
The manuscript presents data from a two year follow up of a sample of adolescents from the Nearby study. The authors have employed objective measurements of sedentary behaviours and physical activity which is a strength, and also some measures of adiposity. This is a timely study and provides insight to developmental changes in these behaviours which is well needed in the literature.
Abstract – it would be useful to include the sample size.
Introduction – Please update reference 3 to reflect the recent publication of global matrix 3.0.
Method:
- Please state whether there differences in baseline measurements between those who participated in follow up and those who did not. You state differences in terms of those who agreed to be followed up but not in terms of those who actually were.
- Please specify if incentives were offered for participation in the follow up.
- Who took the measurements at baseline? Presumably a trained researcher but please include this information.
- The fact that the follow up measures of waist circumference and height and weight are self-reported is certainly a limitation of the study. While this is acknowledged, I can’t help but wonder why this was done, given that researchers went to participants homes to collect the activity monitors and the sample size was so small. However, you can’t retrospectively collect this but I think it would be useful to include some information about any differences that might be expected if these measurement were taken objectively so that readers can understand how much bias there may be.
- Please specify if four valid days for accelerometry included weekend days or if it was any four days.
- It would be good to be consistent with how you specify cut points used and mets for accelerometry – at the moment, sometimes only cut points are specified (i.e. low-LIPA), sometimes just mets (mvpa) and sometimes both (High-LIPA). Overall LIPA could also be clearer by stating 101 counts/minute.
- typo – should be pearson correlation
Results:
- please include standardised effect sizes in table 1.
- figure 1 is not suitable for black and white. Please amend this so it is clearer
Discussion:
- You specify that a diverse sample was included but no such details are given in the manuscript. Further information to support this would be good.
- Currently the conclusion is very brief and does not really address the so what of the study. Please amend to include what the implications of this work are.
Author Response
Dear Reviewer 2:
We greatly appreciate the opportunity to revise and resubmit our manuscript for consideration for publication by Children. We found the feedback thoughtful and constructive, and believe that we have been able to significantly strengthen the manuscript.
Please see the attached the response to the reviewers.
Regards,
Ana María Contardo Ayala, PT MSc.
PhD Candidate
Institute for Physical Activity and Nutrition (IPAN)
School of Exercise and Nutrition Sciences
Deakin University
The manuscript presents data from a two year follow up of a sample of adolescents from the Nearby study. The authors have employed objective measurements of sedentary behaviours and physical activity which is a strength, and also some measures of adiposity. This is a timely study and provides insight to developmental changes in these behaviours which is well needed in the literature
Response: Thank you for your comments on our manuscript.
Abstract:
1. It would be useful to include the sample size.
Response: Thank you for your suggestion. For the journal word-count restrictions we did not include this information. We have reduced some text from the current abstract to include the sample size description:
Lines 19: “Data from 324 and 67 participants were obtained at baseline and follow-up, respectively”
Introduction:
2. Please update reference 3 to reflect the recent publication of global matrix 3.0
Response: This reference has now been updated.
Method:
3. Please state whether there differences in baseline measurements between those who participated in follow up and those who did not. You state differences in terms of those who agreed to be followed up but not in terms of those who actually were.
Response: Thank you for your suggestion, please see our response to Reviewer#1, point#5, and the changes made in the manuscript lines 113-116.
4. Please specify if incentives were offered for participation in the follow up
Response: Compensation was offered to all follow-up participants. This is now specified in the manuscript:
Lines 110-111: “Compensation in the form of a gift voucher was offered to all participants in appreciation of their time”.
5. Who took the measurements at baseline? Presumably a trained researcher but please include this information
Response: This is now clarified in the lines 156-157 as follows:
“At baseline, adolescents’ anthropometric assessments were taken by trained project staff”.
6. The fact that the follow up measures of waist circumference and height and weight are self-reported is certainly a limitation of the study. While this is acknowledged, I can’t help but wonder why this was done, given that researchers went to participants homes to collect the activity monitors and the sample size was so small. However, you can’t retrospectively collect this but I think it would be useful to include some information about any differences that might be expected if these measurement were taken objectively so that readers can understand how much bias there may be.
Response: Thank you for your comment. These health parameters were not collected in the participants’ home because participants were not required to be at home at the time the monitors were collected. The following sentence and respective references were added to the manuscript as follows:
Lines 168-172: “Studies show small discrepancies between objective and self-measured height, weight and waist circumference in adults and young people (e.g. underestimation of weight) [42,43]; self-measured adiposity has been used in epidemiological studies when objective measured are not possible.
7. Please specify if four valid days for accelerometry included weekend days or if it was any four days.
Response: Adolescents were included in the analysis if they had worn the monitor for at least four valid days, which could be week and/or weekend days. This point is clarified as follow:
Lines 144: “Adolescents were included in the analysis if they had worn the monitor for at least four valid days (week and/or weekend days).”
8. It would be good to be consistent with how you specify cut points used and mets for accelerometry – at the moment, sometimes only cut points are specified (i.e. low-LIPA), sometimes just mets (mvpa) and sometimes both (High-LIPA). Overall LIPA could also be clearer by stating 101 counts/minute.
Response: Thank you for this suggestion. The Freedson cut-points utilised are age-specific for MVPA in adolescents, and SED and LIPA cut-point are consistent across all ages. To clarify this we had added an example of 15-year old participant’s cut-point in the text as follow:
Line 150: “…and MVPA (≥4 METS, i.e. for a 15 year old is 2781 counts/minute)”.
9. typo – should be pearson correlation
Response: This is now amended.
Results:
10. please include standardised effect sizes in table 1
Response: Thank you for this suggestion. Cohen’s d has now been added into Table 1.
11. Figure 1 is not suitable for black and white. Please amend this so it is clearer
Response: Figure 1 has now been modified to a black-white format.
Discussion:
12. You specify that a diverse sample was included but no such details are given in the manuscript. Further information to support this would be good
Response: Baseline recruitment was reported previously (reference n=36) and consisted of a sample from a diverse range of high/low walkable communities as well as high/low income areas. This information has now been added:
Lines 90-91: “Baseline recruitment of a diverse sample of adolescents from high/low walkable and income areas has been described previously [36].”
Currently the conclusion is very brief and does not really address the so what of the study. Please amend to include what the implications of this work are.
Response: Thank you for this suggestion. We agree with the reviewer and the conclusion have been modified (within the confines of the journal word limits) as follows:
Lines 290-296: “Changes in average daily sitting and activity intensities and patterns among adolescents followed the direction considered less desirable for health outcomes. However, there were no clear prospective associations with WC, but there was a trend with baseline sitting duration and BMI two years later. In concordance with previous evidence, the findings from this study suggest that adolescents may be at risk of negative health impacts associated with decreases in physical activity and increases in prolonged sitting time over time. Longitudinal research with larger sample sizes and additional assessments points is needed.”
Reviewer 3 Report
This is an excellent addition to the extant knowledge of physical activity and important patterns of sedentary behavior in adolescents. The longitudinal design is a real strength. The challenge of running longitudinal studies is highlighted with only 21% follow-up. But this is normal - it beautifully illustrates just how hard this age group are to access.
One of the real shames is that direct measure of body composition in the 60 follow-up could not have taken place. The authors do explain support was provided to help the youngsters in their self-assessment, it is nonetheless a weakness.
Minor comments: the very first sentence of the introduction is long, and awkward. It would be better broken into two sentences.
The results paragraph would benefit from being split up. The first paragraph could deal with the numbers who wore what. A second paragraph could deal with age,weigh stauts
and a third with the activity/sedentary.
Figure 1 is not easy to follow.
Author Response
Dear Reviewer 3:
We greatly appreciate the opportunity to revise and resubmit our manuscript for consideration for publication by Children. We found the feedback thoughtful and constructive, and believe that we have been able to significantly strengthen the manuscript.
Please see the attached the response to the reviewers.
Regards,
Ana María Contardo Ayala, PT MSc.
PhD Candidate
Institute for Physical Activity and Nutrition (IPAN)
School of Exercise and Nutrition Sciences
Deakin University
Review #3 Comments to the Author:
This is an excellent addition to the extant knowledge of physical activity and important patterns of sedentary behavior in adolescents. The longitudinal design is a real strength. The challenge of running longitudinal studies is highlighted with only 21% follow-up. But this is normal - it beautifully illustrates just how hard this age group are to access.
One of the real shames is that direct measure of body composition in the 60 follow-up could not have taken place. The authors do explain support was provided to help the youngsters in their self-assessment, it is nonetheless a weakness.
Response: Thank you for your comments on our manuscript, please see our response #6 to Reviewer 2. This has now been further addressed in the manuscript.
1. The very first sentence of the introduction is long, and awkward. It would be better broken into two sentences.
Response: The first sentence has now been modified as follows:
Lines 32-35: “Adolescents spend most of their waking hours in sedentary pursuits (i.e. behaviours characterised by low energy expenditure,<1.5 METS, while in a sitting, reclining or lying posture during waking hours (2)). In addition, in Australia, nine in 10 do not meet the minimum recommendation of 60 minutes of moderate-to-vigorous physical activity (MVPA) every day (3).”
2. The results paragraph would benefit from being split up. The first paragraph could deal with the numbers who wore what. A second paragraph could deal with age, weight status and a third with the activity/sedentary.
Response: Thank you for this suggestion, the first paragraph of the results has now been split as suggested.
3. Figure 1 is not easy to follow.
Response: Figure 1 has now been modified
1. Parker KE, Salmon J, Brown HL, Villanueva K, Timperio A. Typologies of adolescent activity related health behaviours. Journal of Science and Medicine in Sport. 2018.
2. Tremblay MS, Aubert S, Barnes JD, Saunders TJ, Carson V, Latimer-Cheung AE, et al. Sedentary Behavior Research Network (SBRN) - Terminology Consensus Project process and outcome. Int J Behav Nutr Phys Act. 2017;14(1):75.
3. Australian Bureau of Statistics 2013. Australian Health Survey: Physical Activity, 2011-2012.